# The supercurrent diode effect and nonreciprocal paraconductivity due to the chiral structure of nanotubes

James Jun He [1,2] ✉, Yukio Tanaka[3] & Naoto Nagaosa [4]

The phenomenon that critical supercurrents along opposite directions become unequal is called the supercurrent diode effect (SDE). It has been observed in various systems and can often be understood by combining spin-orbit coupling and Zeeman field, which break the spatial-inversion and time-reversal symmetries, respectively. Here, we theoretically investigate another mechanism of breaking these symmetries and predict the existence of the SDE in chiral nanotubes without spin-orbit coupling. The symmetries are broken by the chiral structure and a magnetic flux through the tube. With a generalized Ginzburg-Landau theory, we obtain the main features of the SDE in its dependence on system parameters. We further show that the same Ginzburg-Landau free energy leads to another important manifestation of the non-reciprocity in superconducting systems, i.e., the nonreciprocal para-conductivity (NPC) slightly above the transition temperature. Our study suggests a new class of realistic platforms to investigate nonreciprocal prop-erties of superconducting materials. It also provides a theoretical link between the SDE and the NPC, which were often studied separately.

Nonreciprocal transport properties[1] near or inside the super-conducting phase of electronic systems have been attracting a lot of research attention recently. It may manifest itself in nonreciprocal paraconductivity (NPC)[2–8] or in so-called supercurrent diode effect (SDE)[9,10].

In superconductors (SCs) or Josephson junctions with broken inversion ($\mathcal{P}$) and time-reversal ($\mathcal{T}$) symmetries, the critical currents along opposite directions, $J_{c\pm}$, may be unequal, leading to the SDE. This effect has been found in various experimental systems[10–18], part of which may be understood by combining spin-orbit coupling (SOC) and Zeeman field[19–21], which break $\mathcal{P}$ and $\mathcal{T}$, respectively. The SOC-Zeeman mechanism also works in one-dimension[22,23] and in systems with disorders[24]. Supercurrent interferometers may also give rise to the SDE[25–27], where the fractional Josephson effect of Majorana fermions can play a crucial role[25]. There also exist theories that consider sym-metry breakings by internal magnetic[28–32], electric[33,34] or valley[35] orders,

finite momentum pairing[36,37], unconventional superconductivity[38,39], etc. However, systems with magnetic orders may be understood in a way similar to those under Zeeman fields, and superconductors with spontaneous $\mathcal{P}$- or $\mathcal{T}$-breaking pairing are not conveniently found in nature. Thus, it remains an open question whether there exist a new mechanism to generate the SDE in state-of-the-art experimental sys-tems. Finding such a mechanism shall greatly enrich the choice of platforms to investigate the SDE and promote the research in this direction.

While the SDE is a manifestation of a nonreciprocal SC below its transition temperature $T_c$, the nonreciprocity can also be seen slightly above $T_c$, where Cooper pairs start to form but coherent super-conductivity is not reached yet. In this regime, the trend of forming Cooper pairs makes a large contribution to the conductivity, which is called the paraconductivity[40,41]. In systems where $\mathcal{P}$ and $\mathcal{T}$ are broken, the paraconductivity in opposite directions may differ significantly,

[1]Hefei National Laboratory, Hefei, Anhui 230088, China. [2]International Center for Quantum Design of Functional Materials (ICQD), Hefei National Laboratory for Physical Sciences at Microscale, University of Science and Technology of China, Hefei, Anhui 230026, China. [3]Department of Applied Physics, Nagoya University, Nagoya 464-8603, Japan. [4]Center for Emergent Matter Science (CEMS), RIKEN, Wako, Saitama 351-0198, Japan. ✉e-mail: jun_he@ustc.edu.cn

leading to the NPC. Although nonreciprocal conductance may also exist in the normal state at $T \gg T_c$, this effect can be enhanced by several orders of magnitude as the temperature approaches $T_c$[4]. Theories have shown that the NPC can also originate from a combination of SOC and Zeeman field[2,4]. Despite the similarity in the conditions to realize SDE and the NPC, current theories have not discussed the two in the same framework to the best of our knowledge.

In the research works on nonreciprocal transport phenomena in superconductors, both the understanding of current experimental results and the proposals of future platforms focus on systems with magnetization or spin-orbit coupling. A mechanism of generating SDE or NPC in non-magnetic materials without spin-orbit coupling remained elusive.

Here, we reveal such a mechanism with the chiral structure being the key element and predict nanotubes as realistic experimental platforms. We show that both the SDE and the NPC exist in a chiral nanotube under a magnetic field along its axial direction, and they can be obtained with the same generalized Ginzburg-Landau theory. The inversion symmetry is broken by the chiral structure of the nanotube without any SOC, and the magnetic field plays its role through the orbital effect, i.e., Aharonov-Bohm effect, instead of the Zeeman coupling. The resulting nonreciprocal signals strongly depend on the magnetic flux, the nanotube radius, and the chiral angle. There exist a periodicity in the magnetic flux through the tube, similar to the Little-Parks oscillation[42], as well as a periodicity in the chiral angle. The interplay of the magnetic flux and the chiral structure is the origin of both the SDE and the NPC.

The NPC in nanotubes has been observed by Qin et al. in ref. 3 where the nanotubes are formed by transition metal dichalcogenides $WS_2$. A strong SOC exists in this material which may also contribute to the NPC. Our theory is useful to clarify the origin of the observed NPC in ref. 3 and, on the other hand, shows the existence of SDE in chiral structures without SOC. While helping to understand the existing experimental results, this study also serves as a basis for future material choice. Its unified picture of non-reciprocal transport phenomena below and above the superconductivity transition temperature $T_c$ shall be beneficial to the research in both regimes.

## Results

### Chiral nanotubes near $T_c$

A nanotube near its superconductivity transition temperature $T_c$ may be described by the following free energy,

$$F = \int d^2 \mathbf{r} \psi^*(\mathbf{r}) \left[ \alpha + \xi(\hat{\mathbf{p}}) + \frac{\beta}{2} |\psi(\mathbf{r})|^2 \right] \psi(\mathbf{r}), \tag{1}$$

where $\alpha \cdot T - T_c$ and $\beta$ are the conventional Ginzburg-Landau parameters. The displacement vector $\mathbf{r} = (x, y)$ is defined so that the nanotube aligns alone the $x$-direction and the transverse coordinate $y$ circulates around the tube, as illustrated in Fig. 1. The term $\xi(\hat{\mathbf{p}}) = \sum_{ij} \xi_{ij} \hat{p}_x^i \hat{p}_y^j$ is the kinetic energy of a Cooper pair. Apparently, a periodic boundary condition should be applied along the $y$-direction. The momentum operator is $\hat{\mathbf{p}} = -i\hbar\nabla_\mathbf{r} + 2e\mathbf{A}(\mathbf{r})$. Considering a uniform magnetic field applied along the $x$-direction, i.e $\mathbf{H} = H_x\hat{\mathbf{x}}$, and assuming the nanotube wall thickness to be negligible, the vector potential becomes $\mathbf{A} = \frac{\phi}{2\pi R}\hat{\mathbf{y}}$, where $\phi = \pi R^2 H_x$ is the magnetic flux through the nanotube and $R$ is its radius. This is equivalent to a boundary condition $\psi(\mathbf{r}) = \psi(\mathbf{r} + 2\pi R\hat{\mathbf{y}}) \exp\{-2\pi i\phi/\phi_0\}, \phi_0 = h/2e$ being the magnetic flux quantum.

A Fourier transformation (taking into account the magnetic flux) leads to the following equivalent form of Eq. (1),

$$F = 2\pi R \sum_n \int dq \left[ \alpha + \xi(\mathbf{p}) + \frac{\beta}{2}(2\pi R)^2 |\psi_n|^2 \right] |\psi_n|^2, \tag{2}$$

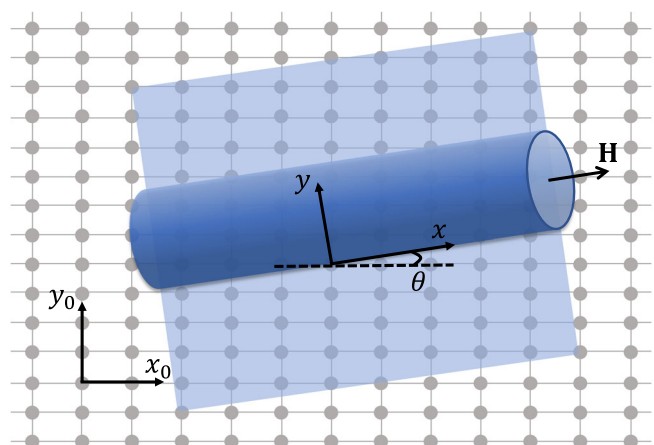

**Fig. 1 | A schematic of a chiral nanotube formed by rolling a two-dimensional sheet.** The two coordinate systems, $(x_0, y_0)$ and $(x, y)$, are connected by a rotation of the chiral angle $\theta$. A magnetic field $\mathbf{H}$ is applied along the tube to generate nonreciprocal effects.

where $q$ is the wavenumber along the tube and $\mathbf{p} = (\hbar q, [n - \phi/\phi_0]\hbar/R)$. The integer $n$ labels the transverse Fourier components. It is quantized due to the small circumference of the tube. We have neglected the coupling between different $\mathbf{q}$-components in the $|\psi|^4$ term, which does not affect the results of this study. It is clear from Eq. (2) that $F$ is a periodic function of $\phi$, leading to the Little-Parks oscillation, as will be seen later.

The chiral structure of the nanotube is reflected in the functional form of $\xi(\mathbf{p})$. To see that, imagine a nano-ribbon obtained by cutting and flattening the nanotube. When the local continuous rotational symmetry ($C_\infty$) of this ribbon is reduced a discrete $C_n$, a chiral nanotube can be obtained if the rolling direction mismatch all the high-symmetry directions. For simplicity, we consider here a system with $C_2$ and the kinetic term may be written as (up to the 4-th order in the momentum)

$$\xi(\mathbf{p}_0) = \frac{|\mathbf{p}_0|^2}{2m_0} + \frac{|\mathbf{p}_0|^4}{4m_0^2\zeta_0} + \frac{p_{x0}^2 - p_{y0}^2}{2m_1} + \frac{\left(p_{x0}^2 - p_{y0}^2\right)^2}{4m_1^2\zeta_1}$$
$$+ \frac{p_{x0}^2\left(p_{x0}^2 - 3p_{y0}^2\right) + p_{y0}^2\left(p_{y0}^2 - 3p_{x0}^2\right)}{4m_2^2\zeta_2} \tag{3}$$

where $\mathbf{p}_0$ is defined in a coordinate system whose axes align with the high-symmetry directions. It is generally different from that of $\mathbf{p}$ defined in the previous coordinate system whose $x$-axis is along the nanotube. They are connected by a rotation of the chiral angle $\theta$, as shown in Fig. 1. The first two terms in Eq. (3) preserves $C_\infty$ while the third term reduces it to $C_2$. Note that $m_1 > m_0$ must hold for the mass along arbitrary direction to be positive. The last two terms are $C_4$ symmetric. The inclusion of quartic terms is necessary to reveal the nonreciprocal properties, similar to the case where such an effect is caused by magnetochiral anisotropy[2,4,20,21].

Equation (3) can be rewritten as

$$\xi(\mathbf{p}) = \frac{p_x^2}{2m_x} + \frac{p_y^2}{2m_y} + \frac{p_x p_y}{m_{xy}} + \sum_{n=0}^{4} \kappa_n p_x^n p_y^{4-n} \tag{4}$$

with, $m_x, m_y, m_{xy}$ and $\kappa_n$ being functions (see Materials and Methods) of the original parameters in Eq. (3). To see how a chiral nanotube breaks $\mathcal{P}$, note that $p_y = (n_y - \phi/\phi_0)\hbar/R$ is defined along a circular coordinate and behaves as angular momentum (rather than the usual momentum in a flat space). It remains unchanged under $\mathcal{P}$ operation, consistent

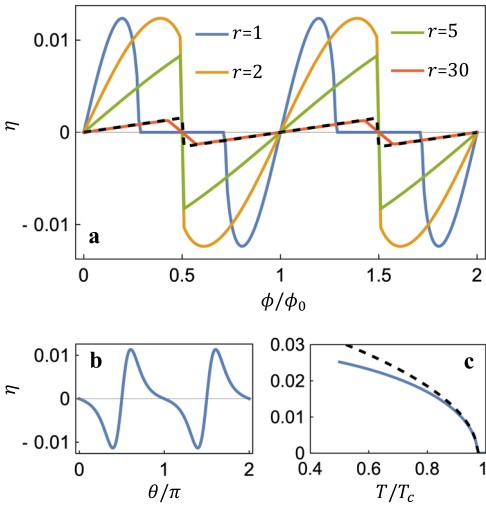

**Fig. 2 | The diode efficiency, $\eta = (J_{c+} - J_{c-})/(J_{c+} + J_{c-})$, obtained by numerically solving for the critical currents $J_{c\pm}$ with Eq. (7). a** The dependence on the magnetic flux ($\phi_0 = h/2e$ is the flux quantum). The solid curves are for various values of the nanotube radius $R$, normalized so that $r = R/l_0$, where $l_0 = \hbar/\sqrt{2m_0 T_c}$. The dashed curve is the approximate result given by Eq. (8) with $r = 30$. **b** Dependence on the angle $\theta$ which corresponds to the chiral structure of the nanotube. **c** The temperature dependence. The parameters are $m_0 = 1$, $m_1 = 2$, $\zeta_2 m_2 \to \infty$, $\zeta_0/T_c = 10$, $\zeta_1/T_c = 20$, $r = 2$, $\theta = 0.6\pi$, $\phi/\phi_0 = 0.3$ and $T/T_c = 0.9$ for all the results unless specified otherwise.

with the symmetry property of the magnetic flux $\phi$ which should not change under spatial inversion. As a result, the nanotube geometry leads to the symmetry operation $(p_x, p_y) \overset{\mathcal{P}}{\to} (-p_x, p_y)$, and thus the $p_x$-odd terms in Eq. (4) break $\mathcal{P}$.

The supercurrent is

$$J_x = -2e \int dy \psi^*(\mathbf{r}) \frac{d\xi}{d\hat{p}_x} \psi(\mathbf{r}) \tag{5}$$

$$= -2e \sum_n \frac{2\pi R}{L} \int dq \frac{\partial \xi(\mathbf{p})}{\partial p_x} |\psi_n(q)|^2, \tag{6}$$

where $L \to \infty$ is the length of the nanotube. With Eqs. (2), (4) and (6), we study the SDE when $T < T_c$ and the NPC when $T > T_c$ in the following.

**Supercurrent diode effect**

When a supercurrent passes through the nanotube, the Cooper pairs acquire a momentum $\mathbf{p}$ and a kinetic energy $\xi(\mathbf{p})$. The order parameter is determined by the Ginzburg-Landau equation as $|\psi_n(q)|^2 = |\alpha|\beta^{-1}(2\pi R)^{-2}(1 - \xi(\mathbf{p})/|\alpha|)$ and the supercurrent is

$$J_x(n, q) = \frac{-2eR}{L^2} \frac{|\alpha|}{\beta R^2} \left(1 - \frac{\xi(\mathbf{p})}{|\alpha|}\right) \frac{\partial \xi(\mathbf{p})}{\partial p_x}. \tag{7}$$

Note that $\alpha < 0$ since $T < T_c$. The critical currents $J_{c\pm}$ are the absolute values of the maximum and minimum, respectively, of $J_x(n, q)$ as $n$ and $q$ are varied.

For general parameters, $J_{c\pm}$ can be determined numerically and the resulting diode efficiency, $\eta \equiv \frac{J_{c+} - J_{c-}}{J_{c+} + J_{c-}}$, is shown in Fig. 2 as functions of the magnetic flux $\phi$, the angle $\theta$ and the temperature, respectively. Figure 2a shows a periodicity in $\phi$, similar to the Little-Parks oscillation. Different curves are for various values of the ratio $r = R/l_0$, with $R$ being the radius of the nanotube and $l_0 = \hbar/\sqrt{2m_0 T_c}$. When $r$ is small and $\phi/\phi_0$ is close to a half-integer, the transverse momentum, $p_y = (n - \phi/\phi_0)\hbar/R \approx \hbar/(2rl_0)$, costs so high a kinetic energy $\xi(\mathbf{p})$ that it

kills the superconductivity (i.e., $\psi_n \to 0$), leading to vanishing $J_{c\pm}$. We define $\eta$ in this case to be zero, resulting in the curve with $r = 1$ in Fig. 2 (a). As $r$ increases, $J_{c\pm}$ becomes nonzero for arbitrary magnetic flux and discontinuities occur as $\phi/\phi_0$ changes across half-integers, which originates from the quantization of the transverse index $n$ in Eq. (7). When $r \gg 1$, discontinuities disappear while non-smooth kinks remain and $|\eta|$ decreases. From Fig. 2b, one finds that $\eta$ vanishes whenever $\theta$ becomes a multiple of $\pi/2$. This is expected because the nanotubes in these cases are not chiral and the inversion symmetry is preserved, forbidding the SDE. As $\theta/\pi$ deviate from half-integers, $|\eta|$ increases sharply and extreme values of $\eta$ are reached quickly. Note that the positions of the extreme points depend on the ratio $m_0/m_1$, which measures the strength (and the sign) of inversion symmetry breaking. The temperature dependence has the usual feature $\eta \sim \sqrt{T_c - T}$, as shown in Fig. 2 (c).

It is helpful to obtain the analytical form of $\eta$, which is possible when $\zeta_{0,1,2} \gg T_c$ and thus the terms with $\kappa_n$ in Eq. (4) can be treated as perturbations. We also assume $r$ to be small, and then varying the transverse quantum number $n$ costs so much energy that $J_{c\pm}$ are obtained with a fixed $n$ in Eq. (7). Under these conditions, the diode efficiency is

$$\eta = \frac{-4}{\sqrt{3}} \left(4\kappa_0 \frac{m_x^2}{m_{xy}} + \kappa_1 m_x\right) m_0 T_c$$
$$\times b \sqrt{\frac{|\alpha|}{T_c} \frac{m_x}{m_0} - b^2 \left(\frac{m_x}{m_y} - \frac{m_x^2}{m_{xy}^2}\right)}, \tag{8}$$

where $b = \phi/\phi_0 - [\phi/\phi_0]$ ([x] denotes the integer closest to $x$). From Eq. (8) it becomes clear that either $m_{xy}^{-1}$ or $\kappa_1$ must be nonzero to achieve the SDE. The requirement, combined with Eqs. (13) and (15), becomes $m_1^{-1} \neq 0$ and $\sin 2\theta \neq 0$, which is just equal to requiring the nanotube to have a chiral structure. When the magnetic filed $H_x$ is small, $\eta$ is linear in $H_x$ (note that $\phi = \pi R^2 H_x$). As the magnetic flux increases, the expression under the square root becomes negative for small $|\alpha|$ since $(\frac{m_x}{m_y} - \frac{m_x^2}{m_{xy}})$ is positive definite. This results in a decrease of the transition temperature to $T_c'$ with $\delta T_c = T_c - T_c' \sim b^2(\frac{m_x}{m_y} - \frac{m_x^2}{m_{xy}})\frac{m_0}{m_x}$. And the temperature dependence of Eq. (8) may be written as $\eta \sim \sqrt{T_c' - T}$. A substitution of Eqs. (12–16) leads to the dashed curves in Fig. 2 (a) and (c), which show great agreement with previous numerical results except two situations, (i) $r \gg 1$ and $\phi/\phi_0$ is close to a half integer and (ii) The temperature is far below $T_c$. In both situations, the assumption that $J_{c\pm}$ can be obtained with the same index $n$ in Eq. (7) no longer holds.

The differences in the SDE between chiral nanotube SCs and previously studied spin-orbit coupled SCs[19–21] is clear now. The diode efficiency here is controlled by the nanotube diameter and the chiral angle, while it is determined by the SOC strength in spin-orbit coupled SCs. The sign change of $\eta$ happens in both kinds of systems as the magnetic field is tuned. However, the origins are rather different. In SOC SCs, $\eta$ changes sign due to the higher-order (in momentum and in field strength) terms in the kinetic energy of the Cooper pairs. Here, it is because the transverse index $n$ corresponding to the critical currents $J_{c\pm}$ is shifted. The sign of $\eta$ changes exactly at $b = 1/2$ here (i.e. when the number of flux quanta is a half-integer) while the sign-flipping field-strength in SOC SCs depends on multiple system parameters.

**Nonreciprocal paraconductivity**

The nonreciprocity of superconducting materials manifests itself not only in the SDE when $T - T_c < 0$, but also in the NPC when $T_c \gg T - T_c > 0$. In the latter case, although the average order parameter vanishes, its quantum fluctuations induce a significant contribution to the conductance, resulting in a drop of resistance above $T_c$ before a

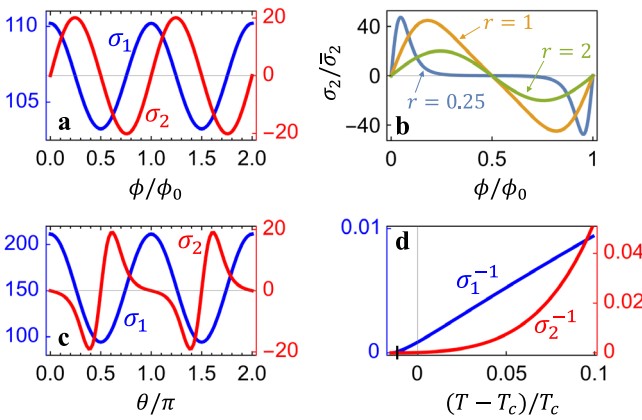

**Fig. 3 | The linear and nonlinear paraconductivity of a chiral nanotube, $\sigma_1$ and $\sigma_2$, normalized by $\overline{\sigma}_1 = \frac{k_B T}{T_c} \frac{e^2}{4\pi^2 \hbar} \frac{\gamma l_0}{R}$ and $\overline{\sigma}_2 = \frac{k_B T}{T_c^2} \frac{e^3}{6\pi^2 \hbar} \frac{\gamma^2 l_0^2}{R}$, respectively. a** Magnetic-flux dependence, showing the Little-Parks oscillation. **b** The evolution of the flux dependence as the normalized radius $r$ is varied. **c** Dependence of $\sigma_{1/2}$ on the chiral angle $\theta$. **d** The temperature dependence of the inverse of $\sigma_{1/2}$. The parameters are the same as those in Fig. 2.

finite order parameter is established. The relation between the two phenomena has not been discussed elsewhere although the symmetry requirements are very similar. In this section, we calculate the para-conductivity of the chiral nanotubes described by Eq. (2) and discuss it in the same framework as we discuss the SDE.

We calculate the paraconductivity using the time-dependent Ginzburg-Landau theory[43] (see Materials and Methods). The resulting current density $j_x = \sigma_1 E + \sigma_2 E^2 + O(E^3)$ where the linear conductivity

$$\sigma_1 = \gamma \frac{T}{T_c} \frac{e^2}{4\pi^2 \hbar} \frac{l_0}{R} \sum_n \int dx \frac{\partial_x^2 f_n(x)}{[\alpha/T_c + f_n(x)]^2}, \tag{9}$$

and the nonreciprocal term

$$\sigma_2 = \gamma^2 \frac{T}{T_c^2} \frac{e^3}{6\pi^2 \hbar} \frac{l_0^2}{R} \sum_n \int dx \frac{\partial_x^3 f_n(x)}{[\alpha/T_c + f_n(x)]^3}. \tag{10}$$

In the dimensionless function $f_n(x) = \xi(\mathbf{p})/T_c$, we made a change of variables, $\mathbf{p} = [p_x, p_y] \rightarrow [x\hbar q_0, y_n \hbar q_0]$, where $y_n = (n - \phi/\phi_0)/R q_0$ and $q_0 = 1/l_0$. The substitution of Eq. (4) leads to

$$f_n(x) = \frac{1}{T_c} \xi(x\hbar q_0, y_n \hbar q_0)$$
$$= \frac{x^2}{2\tilde{m}_x} + \frac{y_n^2}{2\tilde{m}_y} + \frac{xy_n}{\tilde{m}_{xy}} + \sum_{i=0}^4 \tilde{\kappa}_n x^i y_n^{4-i} \tag{11}$$

where $\tilde{m}_{x/y/xy} = m_0^{-1} m_{x/y/xy}$ and $\tilde{\kappa}_i = \kappa_i m_0 (\hbar q_0)^2$ are dimensionless parameters.

The integrals in Eqs. (9) and (10) can be done numerically and the resulting $\sigma_{1/2}$ are shown in Fig. 3 as functions of the magnetic flux $\phi$ and the chiral angle $\theta$. Little-Parks oscillations of both the linear and non-linear conductivities are found in Fig. 3a. The maxima/minima of $\sigma_1$ are at integer/half-integer values of $\phi/\phi_0$ since $\sigma_1$ is an even function of $\phi$ and finite flux suppresses superconductivity. On the hand, the non-reciprocal $\sigma_2$ is odd in $\phi$ and it vanishes whenever $\phi/\phi_0$ becomes a integer. The flux values for optimal $\sigma_2$ depend on the system para-meters such as the nanotube radius, as shown in Fig. 3b. The curves resemble those in Fig. 2a with the difference that they are smooth here because all the transverse components $n \in (-\infty, \infty)$ of the order parameter contribute, unlike the supercurrent which is given by a certain $n$. Fig. 3c shows the effect of the chiral angle $\theta$. The angle dependence of $\sigma_2$ is of similar amplitude to the flux dependence in Fig. 3a. In Fig. 3d, we find that the temperature dependence of $\sigma_1$ is

rather linear, which is similar to higher-dimensional systems[2,4,43]. A difference here is a shifted transition temperature $T_c'$, so that $\sigma_1^{-1} \sim (T - T_c')$. The $T$-dependence of $\sigma_2^{-1}$ is clearly of higher order and we do not find any single power law.

## Discussion

We have shown that superconducting chiral nanotubes with trapped magnetic flux behave as supercurrent diodes, whose diode efficiency strongly depends on the chiral angle. We also found, in the same theoretical framework, that the paraconductivity of such chiral nano-tubes near $T_c$ contains a nonreciprocal part $\sigma_2$, whose dependence on the system parameters is rather similar to that of the SDE efficiency $\eta$ and oscillates periodically as the magnetic flux $\phi$ or the chiral angle $\theta$ is varied. The results show that a combination of inversion symmetry breaking by chiral structure and time-reversal symmetry breaking by magnetic flux can induce nonreciprocal transport properties, includ-ing the SDE and the NPC, in superconductors.

One may notice that actual nanotubes created in laboratories are mostly related to honeycomb or triangular lattices, while the nano-tubes discussed here are obtained by rolling a sheet of rectangular lattice. This choice is for technical convenience. However, the main conclusions drew here shall generally apply. To quantitatively discuss a carbon nanotube (honeycomb) or a transition-metal-dichalcogenide nanotube as experimentally studied in ref. 3 (triangular), terms up to the 6-th order in momentum must be included when constructing their Ginzburg-Landau free energies, which is not really meaningful con-sidering the condition for the validity of the Ginzburg-Landau theory itself. Thus, a study of realistic (carbon/NbSe2/WS2/...) nanotubes may need to use the microscopic BCS theory, which can be done numeri-cally. Another difference of the theory from real materials is that rea-listic nanotubes may be multi-wall and have nonzero thickness, which we ignored here. Our theory is still valid as long as chiral structures are formed and the thickness is much smaller than the superconductivity coherence length. The former condition can be satisfied by sample choice without much difficulty, and the latter one is usually satisfied since the coherence length is quite large in comparison to atomic scales.

Although single superconductors are considered here, the non-reciprocal effects discussed here shall apply to Josephson junctions where two conventional bulk superconductors (Al, Pb, Nb, NbSe2, etc.) are connected by a chiral nanotube. A study of such a system will be of great practical importance. In this manuscript, we aim to clarify the physical principles and general features of the nonreciprocal proper-ties of superconducting chiral nanotubes, and leave more detailed and realistic studies to future works.

Although one needs to break both $\mathcal{P}$ and $\mathcal{T}$ to obtain unequal $J_{c\pm}$[28,44], it should be noted that there also exist nonreciprocal properties in $\mathcal{T}$-preserving Josephson junctions. The nonreciprocity may be observed in unequal retrapping currents $J_{r\pm}$[45] or in ac Josephson effects[9,28]. The interaction between electrons plays an important role in these cases. The design or improvement of supercurrent diodes with strong electron interactions is a topic worth further investigation.

## Methods
### Parameters in the rotated coordinate system
By rotating the coordinate system by the chiral angle $\theta$, one obtains the free energy form in Eq. (4) where the parameters are functions of those in Eq. (3). The functional forms are

$$\frac{1}{m_{x/y}} = \frac{1}{m_0} \pm \frac{\cos 2\theta}{m_1}, \tag{12}$$

$$\frac{1}{m_{xy}} = -\frac{\sin 2\theta}{m_1}, \tag{13}$$

$$\kappa_0 = \kappa_4 = \frac{1}{4}\left(\frac{1}{\zeta_0 m_0^2} + \frac{\cos^2 2\theta}{\zeta_1 m_1^2} + \frac{\cos 4\theta}{\zeta_2 m_2^2}\right), \tag{14}$$

$$\kappa_1 = -\kappa_3 = -\frac{\sin 4\theta}{2}\left(\frac{1}{\zeta_1 m_1^2} + \frac{2}{\zeta_2 m_2^2}\right), \tag{15}$$

$$\kappa_2 = \frac{1}{4}\left(\frac{2}{\zeta_0 m_0^2} + \frac{1 - 3\cos 4\theta}{\zeta_1 m_1^2} - \frac{6\cos 4\theta}{\zeta_2 m_2^2}\right). \tag{16}$$

**Time-dependent Ginzburg-Landau theory**

At a temperature slightly above $T_c$, the fluctuation of the order parameter is determined by the following Langevin equation[43],

$$\hbar\gamma\partial_t\psi(\mathbf{r},t) = -\left[\alpha + \xi(\hat{\mathbf{p}})\right]\psi(\mathbf{r},t) + \delta(\mathbf{r},t), \tag{17}$$

where $\delta(\mathbf{r},t)$ is an uncorrelated random force and $\gamma$ is the inverse of damping constant. Note that $\alpha > 0$ and the static order parameter vanishes, i.e. $\langle\psi_{n,q}(t)\rangle_t = 0$. However, Eq. (17) leads to a nonzero $\langle|\psi_{n,q}(t)|^2\rangle_t$, which is[43]

$$\langle|\psi_{n,q}(t)|^2\rangle = \frac{2k_B T}{\hbar\gamma}\int_{-\infty}^t dt' e^{-\frac{2}{\hbar\gamma}\int_{t'}^t dt''[\alpha + \xi(t'')]}. \tag{18}$$

It is nonzero when an electric field $\mathbf{E} = E\hat{\mathbf{x}}$ is applied, making $\xi(\mathbf{p}(t^*)) = \xi_n(q + 2eEt^*)$. Combining Eqs. (6) and (18), one obtains Eq. (9) and Eq. (10).

## Data availability

All data needed to evaluate the conclusions in the paper are present in the paper.

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

## Acknowledgements

N.N. was supported by JST CREST Grant Number JPMJCR1874, Japan, and JSPS KAKENHI Grant Number 18H03676. Y.T. was supported by Scientific Research (A) (KAKENHI Grant No. JP20H00131), Scientific Research (B) (KAKENHI Grants No. JP20H01857) and JSPS Core-to-Core program Oxide Superspin international network (Grants No. JPJSCCA20170002). J.J.H. was supported by Innovation Program for Quantum Science and Technology (Grant No. 2021ZD0302800).

## Author contributions

N.N. initiated and guided this work. Y.T. helped to analyze the problem. J.J.H. carried out the calculations and wrote the manuscript with suggestions from all the authors.

## Competing interests

The authors declare no competing interests.
