## [Peer Review File · Nature Communications]

The supercurrent diode effect and nonreciprocal paraconductivity due to the chiral structure of nanotubesREVIEWER COMMENTS

Reviewer #1 (Remarks to the Author):

In this manuscript, the authors studied the SC diode effect and nonreciprocal paraconductivity using chiral nanotubes. For the SC diode effect, two key symmetry ingredients are time reversal and inversion. They use the magnetic flux to break the time-reversal symmetry and the chiral structure to break the inversion symmetry. Then, they obtained the SDE and NPC. This work is interesting. But I still have some questions.

- (1) It seems that this work is one extension of the authors' previous work in New. J. Phys. 24, 053014 (2022). In my point of view, the difference here is a modified form of GL free energy from the nanotube. So, why this work is important than the authors' previous work?
- (2) The NPC in chiral nanotube has already been reported and discussed in Ref. [3], Nat. Comm. 8, 14465 (2017). The authors should spend more time and words on this experiment, rather than just cite it in the introduction without discussion.
- (3) One drawback of this setup is the applying magnetic flux. Although we can model this nanotube without thickness, the nanotubes have a finite thickness in realistic cases. I think applying flux along the nanotube may generate vortices. And vortex motion can also cause the supercurrent diode effect, which was found many years ago.

Reviewer #2 (Remarks to the Author):

The supercurrent diode effect is a very active and timely topic of research in superconductivity. The present study proposes to realise the effect in chiral nanotubes. The authors use a generalised Ginzburg-Landau approach to demonstrate the key elements leading to a splitting of the critical supercurrents along the tube. The discussion is then extended in the temperature range above T_c , where non-reciprocity is visible in the paraconductivity.

The proposal is very well substantiated by the theory presented. The theoretical setup is concise, elegant and the explanations are very accessible. The methods used are adequate and can be reproduced based on the presented material. As far as I can judge the proposal is original. This work certainly deserves enhanced attention in view of the originality and the high activity in the field. On the other hand, it is difficult to assess whether it outshines related studies in the field, that it would justify to be published in Nature Communication.

Reply to reviewer #1

Reviewer's Comment:

In this manuscript, the authors studied the SC diode effect and nonreciprocal paraconductivity using chiral nanotubes. For the SC diode effect, two key symmetry ingredients are time reversal and inversion. They use the magnetic flux to break the time-reversal symmetry and the chiral structure to break the inversion symmetry. Then, they obtained the SDE and NPC. This work is interesting. But I still have some questions.

Our reply:

We thank the reviewer for reviewing our manuscript and providing questions, comments, and suggestions. We appreciate that the reviewer found our work interesting. In the following, we reply to the reviewer point by point.

Reviewer's Comment:

(1) It seems that this work is one extension of the authors' previous work in New. J. Phys. 24, 053014 (2022). In my point of view, the difference here is a modified form of GL free energy from the nanotube. So, why this work is important than the authors' previous work?

Our reply:

The Ginzburg-Landau (GL) theory is used in both this manuscript and our previous work mentioned by the reviewer [New. J. Phys. 24, 053014 (2022)]. In our previous work, we proposed to understand the superconductor diode effect (SDE) with a generalized Ginzburg-Landau (GL) theory, focusing on the consequences of breaking inversion and time-reversal symmetries and giving a Rashba spin-orbit coupled superconductor as an example. Breaking the two symmetries is key to the SDE, which had been pointed out in our aforementioned work and in others. However, various methods to break these symmetries to generate SDE in accessible physical systems make a topic that is worth in-depth investigation. Such an investigation is crucial to connect general symmetry arguments to realistic materials and has become an active research subfield during the past year.

The SDE has been observed experimentally in various polar superconductors where the origin of SDE is the interplay between spin-orbit coupling and external Zeeman field (or internal magnetization). Before our work in this manuscript, the understanding of the mechanism of SDE largely relied on spin-orbit coupling, and a mechanism of generating SDE with neither spin-orbit coupling nor electron interaction remained elusive. The motivation of the study in this manuscript is to explore nonreciprocal transport in new superconducting systems. With that in mind, we considered the chiral structures of nanotubes as the source of inversion-symmetry breaking. The form of the GL theory, if no time-dependence was introduced, turned out to have some similarities to previous works, giving the misimpression that it is an extension of previous theories with some terms modified.

However, there are essential differences between this work and our previous one, as we show below. Note that the key points of such theories lie in the physical meanings behind them, which can be very different despite similar mathematical forms. In fact, a generalized GL theory of superconductors that leads to nonreciprocal transport must have terms that break P and T . To the lowest order, there are not many choices of such terms. As a result, such theories will look similar in some ways (especially from mathematical points of view). But the physical consequences can differ a lot.

Moreover, we introduced time dependence in this manuscript to study nonreciprocal paraconductivity (NPC), which is not discussed in the previous work. The NPC itself has been attracting a growing research interest in the past few years. As pointed out in the manuscript, a study of SDE and NPC within the same theoretical framework has not been done before. Such a study is important for a deeper understanding of nonreciprocity in superconducting systems. Drawing the connection between SDE and NPC is also helpful in choosing better materials to achieve each of these effects alone. In addition, the present system has more tuning parameters compared with that in the previous work, which opens a way to design SDE and NPC.

For clarity, we summarize the main differences between this study and our previous work mentioned by the reviewer.

		This manuscript	The previous work
Main differences in the theories	Spatial dimensions	Quasi-one-dimensional, with quantized transverse modes	Fully two-dimensional, with both dimensions continuous
	Time-dependence	Time-dependent GL theory is used to take into account the quantum fluctuations	No time dependence
	Breaking of spatial-inversion symmetry: P	P is broken by the chiral structure of nanotubes;	P breaking is taken into account through the Rashba spin-orbit coupling in polar systems;
	Breaking of time-reversal symmetry: T	T is broken by the flux through the nanotubes, i.e., orbital effect.	T is broken by Zeeman or exchange field
	Preserved symmetries of the theories	Discrete rotational symmetry C_n before rolling the nanoribbons into tubes; No rotational or mirror symmetry after forming tubes	Continuous rotational symmetry C_∞ ; Mirror symmetry $M_{a\hat{x}+b}$ respect to any plane perpendicular to the two-dimensional system.
Main differences in physical consequences	Observables	Critical currents $I_{c\pm}$ & para-conductivity σ	Only $I_{c\pm}$
	Crucial tuning parameters for the SDE efficiency	Magnetic flux, chiral angle, nanotube diameter, effective mass, etc. More tuning parameters.	Zeeman field, chemical potential, etc.
	Magnetic field dependence	Periodic Sign flipping at half-quantized flux values	Not periodic Sign change may or may not happen (depending on detailed system parameters)

One can see that this work and the previous one treat rather different kinds of systems, and different physical quantities are considered.

For the above reasons, we believe our study here is not just an extension of the previous theories, and it has its own importance which should not be undermined by the previous work.

Reviewer's Comment:

(2) The NPC in chiral nanotube has already been reported and discussed in Ref. [3], Nat. Comm. 8, 14465 (2017). The authors should spend more time and words on this experiment, rather than just cite it in the introduction without discussion.

Our reply:

We have added a brief discussion about Ref. [3] in the Introduction and at some other places of the revised manuscript.

Reviewer's Comment:

(3) One drawback of this setup is the applying magnetic flux. Although we can model this nanotube without thickness, the nanotubes have a finite thickness in realistic cases. I think applying flux along the nanotube may generate vortices. And vortex motion can also cause the supercurrent diode effect, which was found many years ago.

Our reply:

We agree that a nanotube can have a finite thickness. However, it is usually small compared to the superconductivity coherence length which determines the size of vortices. As a result, the superconductivity order parameter can be treated as a constant along the thickness (radial) direction and there should be no vortex trapped inside the walls of nanotubes as long as the magnetic field is well aligned along the tube. The magnetic field does lead to a flux trapped inside the tube. It should be noted that this flux is not quantized and is not related to superconductivity vortices. Thus, although vortex motion could indeed contribute to nonreciprocal transport, the magnetic field along the tube does not generate any of them and our results remain valid in realistic nanotubes with finite thickness.

We have added a few sentences in the Discussion of the revised manuscript to discuss the finite thickness of nanotubes.

Reply to reviewer #2

Reviewer's Comment:

The supercurrent diode effect is a very active and timely topic of research in superconductivity. The present study proposes to realise the effect in chiral nanotubes. The authors use a generalised Ginzburg-Landau approach to demonstrate the key elements leading to a splitting of the critical supercurrents along the tube. The discussion is then extended in the temperature range above T_c , where non-reciprocity is visible in the paraconductivity.

Our reply:

We thank the reviewer for reviewing our manuscript and giving professional comments. We agree that the supercurrent diode effect is a very active and timely topic and we found the reviewers' summary of our manuscript accurate.

Reviewer's Comment:

The proposal is very well substantiated by the theory presented. The theoretical setup is concise, elegant and the explanations are very accessible. The methods used are adequate and can be reproduced based on the presented material. As far as I can judge the proposal is original. This work certainly deserves enhanced attention in view of the originality and the high activity in the field.

Our reply:

We are grateful that the reviewer found our proposal well substantiated by our theory and acknowledged the originality, conciseness, elegance, and accessibility of this work.

Reviewer's Comment:

On the other hand, it is difficult to assess whether it outshines related studies in the field, that it would justify to be published in Nature Communication.

Our reply:

The nonreciprocal transport in superconductors, especially the superconductor diode effect (SDE), is a very active research topic, as pointed out by the reviewer in a previous comment, and the numbers of experimental and theoretical works on this problem are rapidly increasing.

However, both the understanding of current experimental results and the proposals of future platforms focus on systems with magnetization or spin-orbit coupling. A mechanism of generating SDE in non-magnetic materials without spin-orbit coupling remained elusive. This study provides such a mechanism with chiral structures being the key element and predicts nanotubes as realistic experimental platforms. Geometric control of SDE/NPC by the chiral structures or nanotubes have not been mentioned in other studies, to the best of our knowledge.

Furthermore, this study provides a theory of the non-reciprocal paraconductivity of superconducting chiral nanotubes, which not only helps to understand the existing experimental results (such as those in Ref. [3]) but also serves as a basis for future material choice. In addition, the present system has more tuning parameters compared with those in other works, which opens a way to design SDE and NPC.

Also, note that we describe the SDE and the NPC in the same framework to get a unified picture of non-reciprocal transport phenomena below and above the superconductivity transition temperature T_c . The connection between these two regimes has not been discussed in other theories. Such a connection shall be beneficial to the research in both regimes.

For the above reasons, our study may point out new directions for future research on the non-reciprocity in superconducting systems, and we believe it qualifies to be published in Nature Communications.

Summary of changes

In the revised manuscript, the main text is edited. The changed parts are in red color.

REVIEWERS' COMMENTS

Reviewer #1 (Remarks to the Author):

The authors have appropriately addressed my questions, I am pleased to recommend this paper for publication.

Reviewer #2 (Remarks to the Author):

The authors have responded to the criticism of both reviewers and made some amendments to the initial manuscript. I stand by my very positive assessment of the first round. I believe that authors have responded well to the comments made by both reviewers and I recommend the publication. The authors argue well on the importance of their research work in the context of other activities in the field.

Reply to Reviewer #1

Reviewer's comment:

The authors have appropriately addressed my questions, I am pleased to recommend this paper for publication.

Our reply:

We are glad that the reviewer found our reply appropriate and thank the reviewer for recommending the publication of this paper.

Reply to Reviewer #2

Reviewer's comment:

The authors have responded to the criticism of both reviewers and made some amendments to the initial manuscript. I stand by my very positive assessment of the first round. I believe that authors have responded well to the comments made by both reviewers and I recommend the publication. The authors argue well on the importance of their research work in the context of other activities in the field.

Our reply:

We are glad that the reviewer found the importance of our research work well-argued. We thank the reviewer for the very positive assessment of our manuscript and for recommending its publication.